# The Relationship between Lifestyle Factors and Obesity Indices among Adolescents in Qatar

**DOI:** 10.3390/ijerph16224428

**Published:** 2019-11-13

**Authors:** Abdelhamid Kerkadi, Abdelmonem H. Sadig, Hiba Bawadi, Al Anoud Mohammed Al Thani, Walaa Al Chetachi, Hammad Akram, Hazzaa M. Al-Hazzaa, Abdulrahman O. Musaiger

**Affiliations:** 1Human Nutrition Department, College of Health Science, QU-Health, Qatar University, Doha 2713, Qatar; monemhassan@gmail.com (A.H.S.); hbawadi@qu.edu.qa (H.B.); 2Ministry of Public Health, Doha 42, Qatar; aaalthani@moph.gov.qa (A.A.M.A.T.); drwalaaa@yahoo.com (W.A.C.); hammadakram77@yahoo.com (H.A.); 3Paediatric Exercise Physiology Research Laboratory, College of Education, King Saud University, Riyadh 11451, Saudi Arabia; alhazzaa@ksu.edu.sa; 4Arab Centre for Nutrition, Manama 26923, Bahrain; amusaiger@gmail.com

**Keywords:** Qatar, adolescents, obesity, cross-sectional study, sedentary behaviour, dietary habits, physical activity

## Abstract

*Background:* Physical inactivity, sedentary behaviour and an unhealthy diet are factors that may increase weight and general and/or abdominal obesity. *Objective:* To evaluate the relationship between general and abdominal obesity and lifestyle factors among adolescents in Qatar. *Methods:* The study data are based on the Arab Teens Lifestyle Study (ATLS). The target population consisted of 1184 adolescents aged between 14 and 18 years old (563 boys and 621 girls), randomly selected through multistage sampling. A validated questionnaire was used to collect data on lifestyle indicators. Anthropometric indicators, which included body weight, height and waist circumference (WC), were measured according to standardised procedures. International Obesity Task Force (IOTF) age- and sex-specific body mass index (BMI) reference values were used to define overweight and obesity. Abdominal obesity was defined by the ‘waist-to-height ratio’ (WHtR > 0.5) and by sex- and age-specific WC cutoff values. *Results:* Females were more inactive than males (63.7% vs. 36.3%; *p* < 0.001). The proportion of adolescents who reported screen time of over 2 h per day was 82.5%. Females engaged in more sedentary behaviour than males (53.4% vs. 46.4%, *p* = 0.009). Being male (OR: 1.3; CI: 1.0–1.7) and skipping breakfast (OR: 1.5; CI: 1.2–2) were significantly associated with overweight/obesity. In contrast, high intake of fast food, fries, sweets and cake were negative predictors of general and abdominal obesity. *Conclusions:* The findings revealed the prevalence of unhealthy lifestyle habits among adolescents in Qatar and indicated relationships between certain dietary habits and obesity. The findings of this study may help in advocating for the implementation of an intervention that includes lifestyle changes targeting adolescents.

## 1. Introduction

In recent decades, Qatar has witnessed a significant improvement in the socioeconomic status of the population, which was accompanied by changes in the lifestyle. These changes were characterised by poor dietary habits, sedentary behaviour and physical inactivity [1]. All these factors have contributed to the rise in non-communicable diseases such as obesity, diabetes and cardiovascular disease [2,3].

Studies conducted in the Arabian Gulf reported a high prevalence of obesity among adults and children, which was considered a public health problem in the region [4,5]. Obesity among children can be associated with an increased risk of adulthood obesity and leads to health problems [6]. In addition to general obesity, abdominal obesity, characterised by a high WHtR and/or high WC, has been associated with an increase in cardiometabolic risk in children and adolescents [7,8].

Different factors have been linked to the development of obesity, which include genetic (genetic susceptibility to obesity), demographic (age, gender, education and income) and lifestyle factors (sedentary behaviour, physical inactivity and poor dietary habits) [9,10]. Evidence from several research studies indicated that lifestyle components such as sedentary behaviour, physical inactivity and dietary habits are the main factors contributing to obesity among children and adolescents [11,12,13,14].

It is well known that physical inactivity is directly associated with weight gain. Results of studies have demonstrated the relationship between physical activity intensity and abdominal obesity among adolescents [15,16]. However, other studies yielded less conclusive results regarding the association between physical activity and abdominal obesity [17,18]. Results of studies conducted to elucidate the relationship between screen time and obesity were inconsistent. Results of some research studies did not find any relationship between screen time and indicators of obesity [19,20,21], while other studies reported an association between sedentary behaviour and abdominal obesity [22,23,24,25]. Dietary habits (i.e., dietary consumption patterns, meal frequency, skipping breakfast) have been considered key factors contributing to obesity among children and adolescents. Several studies have reported high prevalence of obesity among children and adolescents adopting a diet composed of unhealthy foods such as fast food, sweets and sweetened beverages [13,26]. Other studies indicated that the frequency of meal intake may contribute to the development of obesity [16,27]. To our knowledge, there is a lack of data on the association between lifestyle factors and general and/or abdominal obesity among adolescents in Qatar. Therefore, the objective of the study was to evaluate the relationship between general and abdominal obesity and lifestyle factors among adolescents in Qatar.

## 2. Materials and Methods

The study data are based on the Arab Teens Lifestyle Study (ATLS). The ATLS is a school-based, cross-sectional and multi-centre collaborative study. It was conducted in many Arabic countries to evaluate lifestyle factors associated with obesity among Arab adolescents. The objectives, design and methods of the ATLS have been described before [28].

### 2.1. Study Population

The target population of the present study was adolescents (boys and girls) enrolled in independent schools in Qatar during the academic year 2013/2014. Details of the sampling, selection methods, inclusion and exclusion criteria have been published elsewhere [29]. In the first stage of sampling, 47 schools were listed and numbered. Every other school was selected, with the first number being chosen randomly. A total of 23 schools were selected (11 boys’ school and 12 girls’ schools). In the second stage of sampling, one class from each of grade 10, grade 11 and grade 12 was randomly selected.

The total number of adolescents selected to participate in the study was 1184, consisting of 563 boys and 621 girls. Of these, only 1161 were included in the analysis. The review committee of the Ministry of Public Health and the Ministry of Education and Higher Education (Department of Policy Analysis and Research) approved the protocol for the study. Consent for participation in the study was obtained from school principals. Students also consented to participate in the study.

### 2.2. Anthropometric Measurements

Anthropometric indicators, which included body weight, height and WC, were measured according to standardised procedures. BMI was calculated as body weight in kilogrammes divided by height in square metres. Overweight and obesity were defined according to the IOTF age- and sex-specific BMI reference values [30]. WHtR was calculated as the ratio between WC in centimetres and height in centimetres. Abdominal obesity was defined by using the WHtR cutoff point of 0.50 [31] and the WC cutoff adopted by Taylor et al. [32].

### 2.3. Research Instrument and Data Collection

Details of the validity and reliability of the ATLS questionnaire were published earlier. The questionnaire has shown high reliability (*r* = 0.87) and acceptable validity (*r* = 0.37, *p <* 0.001) [33,34].

### 2.4. Physical Activity Assessment

Time spent on physical activity was collected using a validated self-reported questionnaire [33]. Each participant was asked to report the duration, intensity and frequency of any physical activity from the list practised during the week. The list provided included different physical activities: household activities, recreational activities and sport activities. The metabolic equivalent values were used to differentiate between different levels of physical activities (light, moderate and vigorous) [35,36]. Furthermore, total metabolic equivalent (METs-min/week) was split into active and inactive according to a cutoff score of 1680 METs-min/week, which is the equivalent of one hour of daily moderate-intensity activity (60 min × 7 days/week × 4 METs) [37].

### 2.5. Sedentary Behaviour

Sedentary behaviour included information on time spent during the day in front of a screen watching TV, playing video games and using a computer. The cutoff point for total screen time was set to a maximum of 2 h per day, in accordance with the American Academy of Pediatrics guidelines [38].

### 2.6. Dietary Habits

The dietary habits of adolescents were determined through a set of 10 specific questions. Adolescents were asked to report on the frequency of intake of breakfast, sugar-sweetened drinks (including soda), cooked and uncooked vegetables, fruits, milk and dairy products, doughnuts/cakes, candy and chocolate, energy drinks, and Western and Arabic fast foods in a typical week.

The frequency scales ranged from zero intake (never) to a maximum intake of seven days per week (every day). Intake levels were divided into three groups: low (1–2 days per week); medium (3–4 days per week) and high (>5 days per week). The proportion of adolescents who consumed healthy foods (breakfast, fruits, vegetables and milk) daily and those whose intake per week of unhealthy foods (fries, sweets, fast food, candy) exceeded four days was determined.

### 2.7. Statistical Analysis

The data were analysed using IBM SPSS statistics for Windows, version 23, Chicago, USA. Data were presented as the mean, standard deviation and proportion. The student’s *t*-test and ƛ^2^ test were used to assess differences between genders. The relationship between general obesity and/or abdominal obesity and lifestyle components was determined by using multivariate logistic regression. Moreover, *p* < 0.05 was considered statistically significant.

## 3. Results

Table 1 presents the anthropometric indicators of the study population. In total, 1161 adolescents with complete data were included in the analysis; 51.7% were females (*N* = 600) and 48.3% were males (*N* = 561). The average age was 16.3 ± 1.0. Statistically significant weight, height, WC and WHtR differences were found between genders (*p* < 0001). There were no differences in BMI between males and females. The prevalence of general and abdominal obesity was 41.3% and 46.9% respectively. The prevalence of general obesity was significantly higher in males (44.6%) than in females (38.2%). The same pattern was observed for abdominal obesity, where 50.4% and 43.5% of males and females had abdominal obesity respectively. The prevalence of abdominal obesity was statistically higher for males than for females.

Table 2 shows sedentary behaviour, physical activity and dietary habits according to gender. Results indicated that 44% of adolescents were inactive. Females were significantly inactive compared to males (63.7% vs. 36.3%; *p <* 0.001). Regarding screen time, 82.5% of adolescents reported a screen time of more than 2 h per day. The rate of highly sedentary behaviour (more than 2 h per day) was higher for females than for males (53.4% vs. 46.6%, *p =* 0.009). Moreover, 20.3% and 13.9% of adolescents reported regularly consuming vegetables and fruits, respectively. There were no significant differences in the consumption of vegetables and fruits between males and females. The rate of daily milk consumption was 24.1%. Males were more likely than females to consume milk regularly (55.9% vs. 44.1%, *p <* 0.01). Furthermore, 48.8% and 27.3% of adolescents regularly consumed sweetened beverages and fast food, respectively. There were no significant differences between the genders in relation to the weekly reported intake of sweetened beverages and fast food. Half of the adolescents reported consuming candy and chocolate on at least four days per week. The prevalence was higher for females than for males (60.2% vs. 39.8%, *p <* 0.001). In addition, 28.7% and 24.5% of participants consumed fries and cake on at least four days per week, respectively. Statistically significant differences (*p <* 0.05) were found for the consumption of cake and fries between males and females. Regular consumption of energy drinks was more prevalent among males than females (62.4% vs. 37.6%, *p <* 0.001).

Lifestyle variables according to BMI, WHtR and WC are presented in Table 3 and Table 4. The results indicated statistically significant differences (*p <* 0.05) for lifestyle indicators between males and females. Males were more active and less sedentary and adopted healthy diet patterns, characterised by more frequent consumption of breakfast, vegetables, fruits and milk, whereas females were more sedentary and less active. Females adopted unhealthy diet choices, with more frequent consumption of fast food, fries, candy and sweets. Males with general obesity and abdominal obesity were significantly less active (*p <* 0.001) and reported significantly less frequent consumption of breakfast (*p <* 0.05). Males with normal BMI, WHtR or WC more frequently consumed fast food and candy. Energy drink intake was significantly higher (*p <* 0.05) among males with normal WHtR and/or WC (see Table 3). Obese females reported significantly less frequent consumption of breakfast (*p <* 0.05), fries (*p <* 0.05), cake (*p <* 0.001) and candy (*p <* 0.05). The same patterns were observed for females with high WHtR and/or WC.

Univariate associations between general obesity and AO according to lifestyle components are shown in Table 5. The results demonstrated that being male (OR: 1.3; CI: 1.0–1.7) and not having breakfast (OR: 1.5; CI: 1.2–2) were associated with a significantly higher risk (*p <* 0.05) of the development of overweight/obesity. Males had a significantly higher risk of being abdominally obese (elevated WHtR (OR: 1.3; CI: 1.1–1.7) and/or having elevated WC (OR: 1.7; CI: 1.4–2.2]). In addition, the results indicated that a high consumption of fast food, fries, sweets and cake was negatively associated with overweight/obesity and abdominal obesity.

## 4. Discussion

The present study documented lifestyle factors associated with general and abdominal obesity among adolescents in Qatar. The results of the study indicated a high prevalence of general and abdominal obesity among the study population. The rates of overweight and obesity were higher than those reported from several other countries, including Sweden (15%) [39], Portugal (29%) [40], Spain (21%) [41], Lebanon (32.2%) [42], Iraq (25.3%) [43] and Sudan (10.6%) [44]. The results of the present study demonstrated a high prevalence of abdominal obesity. The prevalence of abdominal obesity, indicated by elevated WHtR, was higher than that reported in Lebanon (20.9%) [45], Saudi Arabia (33.1%) [46] and Portugal (22.5%) [40], but lower than that in Jordan (49.4%) [47]. Rates of elevated WC (48.8%) were higher than those found in Malaysia (11.3%) [48], Lebanon (14.1%) [45] and Brazil (5%) [49]. We noted a statistically significant difference between genders in the prevalence of general and abdominal obesity, with the risk of obesity being higher in males than in females. Similar results were reported in other countries such as Sweden [39]. Adolescent males were almost twice as likely to be abdominally obese than females based on the WHtR indicator. The findings of the current study corroborate those reported in Saudi Arabia [44] and Lebanon [41,44].

Childhood obesity has been associated with biological, behavioural and environmental determinants. Numerous research studies have illustrated the role lifestyle factors (physical inactivity, poor diet, insufficient sleep, and excessive sedentary behaviour) play in the obesity epidemic in children [50,51,52]. Physical activity and sedentary time are considered to be the greatest contributors to obesity and the health risks associated with it [53].

The results of the present study revealed that 82.5% of adolescents exceeded the recommended cutoff for screen time (2 h/day). The results of the present study align with those from Kuwait [27] and Saudi Arabia [16]. However, this rate was higher than that reported by other studies in Qatar [54], the United States [55], Jordan [47], Sudan [44], Lebanon [42], Sweden [39] and Morocco [56]. We found that females were more involved in sedentary behaviour than males. Other cross-sectional studies have shown similar results [16,27,39,54]. However, other studies did not find any significant differences between genders [43,44,46,54,55].

The results of studies investigating the association between sedentary behaviour and obesity were contradictory. We did not find any relationship between general and/or abdominal obesity and screen time. This research result is consistent with other studies [16,17,21,27,39,47,48]. However, other studies have noted a significant association between adiposity and screen time [23,24,42,45]. The consumption of an unhealthy diet, especially sugary snacks, while watching TV may be considered a contributing factor. Other researchers have reported an association between energy-dense sugary snacks and screen time [13,57,58].

Studies have reported a high prevalence of physical inactivity among adolescents [59]. The results for physical activity have shown that only 56% of adolescents in Qatar were physically active. We found differences in reported physical activity between genders (*p <* 0.001). The proportion of physically active males was higher than that of females (66.9% vs. 45.8%). Similar results have been reported by other studies [28,42,54,60]. In a comparison study including seven Arabic countries, Guthold et al. found that 85% and 75% of males and females, respectively, aged 13 to 15 years old were inactive [61]. The difference in physical activity between the genders may be explained by cultural aspects that do not encourage girls to participate in physical activities. Among the perceived barriers to physical activity for females are inaccessibility to sport facilities, inability to exercise outdoors in sportswear and a lack of parental support [62,63].

While growing evidence has emphasised the role of physical inactivity in the development of obesity among children and adolescents [17,39,40,42], the present study did not find any significant association between total physical activity and general or abdominal obesity. The results of the study are consistent with those of other studies [17,28,44,46,49].

The results on dietary habits revealed that overweight/obesity was significantly associated with a low frequency of breakfast intake. A similar association was reported by other studies [16,27,40,64,65,66]. The results from the International Study of Childhood Obesity, Lifestyle and the Environment indicated an inverse relationship between BMI-z scores and body fat percentage and frequency of breakfast intake [67]. The results of a longitudinal Australian study demonstrated a high risk of overweight/obesity among children and adolescents skipping breakfast [65]. However, other studies did not report any relationship between breakfast intake and overweight/obesity and/or abdominal obesity [11,45]. The protective effect of breakfast intake on obesity and overweight may be associated with low snacking and the consumption of high-calorie foods during the day [68,69]. In the present study, adolescents with general and/or abdominal obesity reported a low frequency of consumption of unhealthy foods such as fast food, fries, cake, sugar-sweetened beverages and sweets. In addition, a significant inverse association was documented between higher intake of unhealthy foods and a lower risk of overweight/obesity and abdominal obesity. Our results align with those reported in Saudi Arabia [16] and Portugal [41]. Many studies found a link between the consumption of unhealthy foods and general and/or abdominal obesity [27,44,70,71]. The negative association between consumption of unhealthy foods and obesity could be due to the fact that obese people under-report their intake. In addition, they pay more attention to foods with high energy density [72].

The current study has some limitations. The data are drawn from a cross-sectional study; hence, it was not possible to draw inferences about causality between the study variables and obesity. An additional limitation is the use of a self-administered instrument to collect lifestyle indicators, which may lead to errors. Moreover, the food frequency questionnaire did not include portion sizes. The strengths of the study include the sample size and the use of a validated questionnaire.

## 5. Conclusions

The present study revealed significant associations between low frequency of intake of breakfast, sugar-sweetened beverages, fast food, fries and sweets and obesity, in addition to low physical activity and high sedentary behaviour among adolescents. The findings of this study will help with developing an intervention that will focus on promoting physical activity and healthy eating in schools.

## Figures and Tables

**Table 1 ijerph-16-04428-t001:** Anthropometric indicators of the study population.

Variable	Male	Female	Total
Number of participants	561	600	1161
Age (year)	16.3 ± 0.9	16.3 ± 1.1	16.3 ± 1.0
Weight (kg) ^e^	71.2 ± 21.5	60.4 ± 17.6	65.6 ± 20.3
Height (cm) ^e^	169.9 ± 7.3	157.6 ± 5.8	163.5 ± 9.0
BMI (kg/m^2^)	24.5 ± 6.8	24.2 ± 6.6	24.4 ± 6.7
WC (cm) ^e^	88.9 ± 16.0	79.5 ± 14.5	84.0 ± 15.9
WHtR ^e^	0.52 ± 0.09	0.50 ± 0.09	0.51 ± 0.51
Overweight and obesity ^a,d^ (%)	44.6	38.2	41.3
AO ^b,d^ (%)	50.4	43.5	46.9
AO ^c,d^ (%)	55.8	42.2	48.8

Data presented as means ± standard deviation for continuous variables and % for categorical variables; BMI = body mass index; WC = waist circumference; WHtR = waist-to-height ratio; ^a^ Overweight and obesity defined by IOTF standard (Cole et al., 2000); AO = abdominal obesity; ^b^ WHtR > 0.5; ^c^ Waist circumference cutoff adopted by Taylor et al. (2000); ^d^
*p <* 0.05; ^e^
*p <* 0.001. *t*-test (continuous variables) and ƛ^2^ test t (categorical variables).

**Table 2 ijerph-16-04428-t002:** Physical activity, sedentary behaviour and dietary habits by gender.

Variable	Total	Male	Female	*p*-Value
	*N* = 1161 (%)	*N* = 561 (%)	*N* = 600 (%)	
Total screen time (TV viewing and computer use) *				**0.009**
Less than 2 h/day	203 (17.5)	115 (20.5)	88 (14.7)	
2 h or more/day	958 (82.5)	446 (79.5)	512 (85.3)	
Physical activity				**<0.001**
Inactive	510 (44)	185 (33.1)	325 (54.2)	
Active	649 (56)	374 (66.9)	275 (45.8)	
Sleeping pattern				0.188
Less than 8 h/day	785 (68)	389 (9.8)	396 (66.2)	
8 h or more/day	370 (32)	168 (30.2)	202 (33.8)	
Breakfast intake				0.283
Daily	310 (26.8)	158 (28.2)	152 (25.4)	
Less than daily	848 (73.2)	402 (71.8)	446 (74.6)	
Consumption of vegetables				
Daily	235 (20.3)	112 (20.1)	123 (20.5)	0.845
Less than daily	922 (79.7)	446 (79.9)	473 (79.5)	
Consumption of fruits				0.089
Daily	161 (13.9)	88 (15.7)	73 (12.2)	
Less than daily	995 (86.1)	472 (84.3)	523 (87.8)	
Consumption of milk and dairy products				**0.003**
Daily	279 (24.1)	156 (27.9)	123 (20.5)	
Less than daily	879 (75.9)	403 (72.1)	476 (79.5)	
Intake of sugar-sweetened beverages				0.073
Fewer than 4 days/week	593 (51.2)	271 (48.5)	322 (3.8)	
4 days or more/week	565 (48.8)	288 (1.5)	277 (46)	
Fast food intake				0.053
Fewer than 4 days/week	843 (72.7)	422 (75.4)	421 (70.3)	
4 days or more/week	316 (27.3)	138 (24.6)	178 (29.7)	
Intake of fries/crisps				**0.034**
Fewer than 4 days/week	826 (71.3)	415 (74.2)	411 (68.6)	
4 days or more/week	332 (28.7)	144 (25.8)	188 (31.4)	
Cake/doughnut intake				**0.006**
Fewer than 4 days/week	874 (75.5)	442 (79.1)	432 (72.1)	
4 days or more/week	284 (24.5)	117 (20.9)	167 (27.9)	
Candy/chocolate intake				**<0.001**
Fewer than 4 days/week	585 (50.6)	331 (59.3)	254 (42.5)	
4 days or more/week	571 (49.4)	227 (40.7)	344 (57.5)	
Energy drink intake				**<0.001**
Fewer than 4 days/week	963 (83.2)	437 (78.3)	526 (87.8)	
4 days or more/week	194 (16.8)	121 (21.7)	73 (12.2)	

* Including both weekdays and weekends. A ƛ^2^ test was used to compare between genders. Bold indicated significant differences.

**Table 3 ijerph-16-04428-t003:** Lifestyle variables according to BMI, WHtR and WC for adolescent males in study population.

Variable	BMI ^a^		WHtR		WC ^b^	
	NW (*N* = 311)	OW+OB (*N* = 250)	WHtR <0.5 (*N* = 278)	WHtR ≥0.5 (*N* = 283)	Normal WC (*N* = 248)	Elevated WC (*N* = 313)
Total screen time (TV viewing and computer use) (h/day)	4.82 ± 3.13	4.98 ± 3.30	4.82 ± 3.13	4.96 ± 3.23	4.72 ± 3.09	5.02 ± 3.28
METs-min/week	4021.6 ± 3166.5 *	3236.8 ± 2674.8	4196.0 ± 3281.9 **	3146.5 ±2735.5	4301.3 ± 3281.9	3167.4 ±2792.7
METs-min/week from vigorous activity	2847.7 ± 2611.6 **	2107.4 ± 2148.3	2995.5 ± 2710.1 **	2048.1 ± 2044.2	3034.9 ± 2676.3 **	2111.0 ± 2159.5
METs-min/week from moderate activity	1115.6 ± 1034.0	1091.0 ± 1141.8	1142.3 ± 1040.4	1067.4 ± 1122.8	1196.8 ± 1072.3 **	1031.6 ± 1086.6
Average sleep time per day (h/day)	6.94 ± 1.76	6.76 ± 1.86	6.91 ± 1.77	6.81 ± 1.85	6.96 ± 1.787	6.78 ± 1.83
Breakfast intake (frequency/week)	3.92 ± 2.62 *	3.41 ± 2.60	3.93 ± 2.58 *	3.46 ± 2.65	3.93 ± 2.58 *	3.51 ± 2.65
Vegetable consumption (frequency/week)	3.93 ± 2.20	3.93 ± 2.19	3.87 ± 2.18	4.01 ± 2.21	3.95 ± 2.12	3.92 ± 2.21
Fruit consumption (frequency/week	3.70 ± 2.18	3.46 ± 2.15	3.66 ± 2.18	3.52 ± 2.16	3.62 ± 2.19	3.57 ± 2.16
Milk/dairy products (frequency/week)	4.08 ± 2.41	3.94 ± 2.44	3.95 ± 2.40	4.09 ± 2.44	3.95 ± 2.37	4.08 ± 2.46
Sugar-sweetened beverages (frequency/week)	4.02 ± 2.32	3.64 ± 2.42	4.01 ± 2.34	3.69 ± 2.40	3.95 ± 2.34	3.77 ± 2.40
Fast food intake (frequency/week)	2.69 ± 1.96 *	2.35 ± 1.70	2.71 ± 1.97 *	2.37 ± 1.73	2.77 ± 2.01 *	2.36 ± 1.71
Fries/crisps intake (frequency/week)	2.59 ± 1.95	2.44 ± 1.80	2.64 ± 1.93	2.42 ± 1.84	2.57 ± 1.95	2.50 ± 1.84
Cake/doughnut intake (frequency/week)	2.41 ± 1.87	2.23 ± 1.94	2.54 ± 1.89 *	2.13 ± 1.90	2.42 ± 1.86	2.26 ± 1.95
Sweets/candy intake (frequency/week)	3.58 ± 2.19 *	2.94 ± 2.10	3.58 ± 2.23 *	3.01 ± 2.08	3.51 ± 2.24 *	3.12 ± 2.11
Energy drink intake (frequency/week)	1.87 ± 2.41	1.58 ± 2.20	1.94 ± 2.43 *	1.54 ± 2.19	2.05 ± 2.46 *	1.50 ± 2.17

Data are means ± standard deviation; BMI = body mass index; NW = normal, OW = overweight; OB = obese; WC = waist circumference; WHtR = waist-to-height ratio. ^a^ Overweight and obesity defined by IOTF standard (Cole et al., 2000); ^b^ Waist circumference cut-off adopted by Taylor et al. (2000); ^a^
*p <* 0.05; ** *p <* 0.001. The student’s *t*-test was used to compare groups within each obesity indicator.

**Table 4 ijerph-16-04428-t004:** Lifestyle variables according to BMI, WHtR and WC for adolescent females in study population.

Variable	BMI ^a^		WHtR		WC ^b^	
	NW (*N* = 371)	OW+OB (*N* = 229)	WHtR < 0.5 (*N* = 339)	WHtR ≥ 0.5 (*N* = 261)	Normal WC (*N* = 347)	Elevated WC (*N* = 253)
Total screen time (TV viewing and computer use) (h/day)	5.99 ± 3.75	6.01 ± 3.70	6.01 ± 3.71	5.99 ± 3.76	5.95 ± 3.67	6.07 ± 3.78
METs-min/week	2240.9 ± 2254.4	2140.4 ± 1843.6	2275.4 ± 2236.1	2108.0 ± 1924.3	2314.7 ± 2285.7	2048.8 ± 1824.6
METs-min/week from vigorous activity	863.5 ± 1214.9	767.8 ± 842.3	847.5 ± 1194.1	800.2 ± 934.4	861.7 ± 1211.2	779.3 ± 892.2
METs-min/week from moderate activity	794.6 ± 927.1	751.4 ± 817.6	798.3 ± 945.6	759.4 ± 804.5	801.1 ± 934.9	746.6 ± 816.0
Average sleep time per day (h/day)	6.89 ± 1.94	6.78 ± 2.01	6.86 ± 1.99	6.84 ± 1.95	6.87 ± 1.98	6.82 ± 1.96
Breakfast intake (frequency/week)	3.61 ± 2.58 *	2.96 ± 2.53	3.53 ± 2.52	3.16 ± 2.65	3.44 ± 2.52	3.26 ± 2.66
Vegetable consumption (frequency/week)	3.63 ± 2.36	3.55 ± 2.35	3.55 ± 2.39	3.66 ± 2.31	3.56 ± 2.39	3.66 ± 2.30
Fruit consumption (frequency/week)	3.04 ± 2.09	3.02 ± 2.17	3.01 ± 2.09	3.07 ± 2.16	3.00 ± 2.10	3.09 ± 2.15
Milk/dairy products (frequency/week)	3.50 ± 2.45	3.49 ± 2.40	3.45 ± 2.39	3.50 ± 2.45	3.40 ± 2.41	3.62 ± 2.46
Sugar-sweetened beverages (frequency/week)	3.86 ± 2.30	3.47 ± 2.36	3.93 ± 2.30 *	3.43 ± 2.34	3.89 ± 2.30 *	3.46 ± 2.36
Fast food intake (frequency/week)	2.90 ± 1.97	2.71 ± 1.89	3.01 ± 1.99 *	2.59 ± 1.85	3.01 ± 2.00 *	2.57 ± 1.83
Fries/crisps intake (frequency/week)	3.12 ± 2.00 *	2.61 ± 1.94	3.18 ± 2.03 **	2.60 ± 1.90	3.19 ± 2.03 **	2.56 ± 1.89
Cake/doughnut intake (frequency/week)	3.05 ± 2.01 **	2.32 ± 1.75	3.06 ± 2.01 **	2.40 ± 1.80	3.00 ± 1.98 *	2.46 ± 1.86
Sweets/candy intake (frequency/week)	4.35 ± 2.15 *	3.86 ± 2.10	4.37 ± 2.15 *	3.90 ± 2.11	4.34 ± 2.14 *	3.93 ± 2.13
Energy drink intake (frequency/week)	1.02 ± 1.82	1.14 ± 1.95	1.09 ± 1.86	1.05 ± 1.88	1.05 ± 1.84	1.09 ± 1.91

Data are means ± standard deviation; BMI = body mass index; NW = normal, OW = overweight; OB = obese; WHtR = waist-to-height ratio; WC = waist circumference. ^a^ Overweight and obesity defined by IOTF standard (Cole et al., 2000); ^b^ Waist circumference cutoff adopted by Taylor et al. (2000); * *p <* 0.05; ** *p <* 0.001. The student’s *t*-test was used to compare groups within each obesity indicator.

**Table 5 ijerph-16-04428-t005:** Association between lifestyle risk factors with general and abdominal obesity.

Variables	Overweight/Obesity ^a^			Abdominal Obesity WC ^b^			Abdominal Obesity WHtR ^c^		
	OR	95% CI	*p*-Value	OR	95% CI	*p*-Value	OR	95% CI	*p*-Value
**Sex**									
Male	1.3	1.0–1.7	**0.024**	1.7	1.4–2.2	**>0.001**	1.3	1.1–1.7	**0.017**
Female	*Reference*			*Reference*			*Reference*		
**Physical activity** ^d^									
Inactive	0.996	0.8–1.3	0.970	1.1	0.9–1.4	0.273	1.1		
Active	*Reference*			*Reference*			*Reference*	0.9–1.4	0.354
**Screen time**									
≤2 h/day	1.0	0.7–1.4	0.982	0.9	0.7–1.2	0.589	0.9	0.7–1.3	0.718
<2 h/day	*Reference*			*Reference*			*Reference*		
**Sleep category**									
<8 h/day	1.2	0.9–1.5	0.230	1.2	0.9–1.6	0.132	1.1	0.9–1.4	0.358
≥8 h/day	*Reference*			*Reference*			*Reference*		
**Breakfast intake**									
1–2 days/week	1.5	1.2–2.0	**0.002**	1.1	0.9–1.5	0.328	1.3	1.0–1.6	0.060
3–4 days/week	1.4	1.0–2.0	0.090	1.0	0.7–1.4	0.871	1.0	0.7–1.4	0.998
≥5 days/week	*Reference*			*Reference*			*Reference*		
**Vegetable intake**									
1–2 days/week	1.1	0.9–1.5	0.409	1.0	0.8–1.3	0.944	0.9	0.7–1.2	0.512
3–4 days/week	1.1	0.8–1.5	0.586	0.9	0.6–1.2	0.390	0.8	0.6–1.1	0.248
≥5 days/week	*Reference*			*Reference*			*Reference*		
**Fruit intake**									
1–2 days/week	1.0	0.8–1.4	0.763	0.9	0.7–1.2	0.366	0.9	0.7–1.2	0.606
3–4 days/week	0.9	0.6–1.2	0.431	0.9	0.7–1.2	0.542	0.8	0.6–1.1	0.258
≥5 days/week	*Reference*			*Reference*			*Reference*		
**Milk and milk products intake**									
1–2 days/week	1.1	0.8–1.4	0.560	0.9	0.7–1.1	0.262	0.9	0.7–1.2	0.509
3–4 days/week	1.1	0.8–1.5	0.695	0.7	0.5–1.1	0.064	0.8	0.6–1.1	0.129
≥5 days/week	*Reference*			*Reference*			*Reference*		
**Sugar-sweetened beverage intake**									
1–2 days/week	*Reference*			*Reference*			*Reference*		
3–4 days/week	0.9	0.7–1.2	0.510	0.8	0.6–1.1	0.224	0.8	0.6–1.2	0.283
≥5 days/week	0.7	0.5–0.9	0.10	0.8	0.6–1.0	**0.037**	0.7	0.5–0.9	**0.008**
**Fast food intake**									
1–2 days/week	*Reference*			*Reference*			*Reference*		
3–4 days/week	0.9	0.6–1.1	0.277	0.8	0.6–1.1	0.129	0.8	0.6–1.1	0.148
≥5 days/week	0.6	0.5–0.9	**0.006**	0.5	0.4–0.7	**>0.001**	0.6	0.4–0.8	**<0.001**
**Sweets/candy intake**									
1–2 days/week	*Reference*			*Reference*			*Reference*		
3–4 days/week	0.8	0.6–1.1	0.277	0.9	0.6–1.2	0.393	0.9	0.7–1.2	0.522
≥5 days/week	0.6	0.4–0.8	**>0.001**	0.6	0.5–0.8	**>0.001**	0.7	0.6–0.4	**>0.001**
**French fries Intake**									
1–2 days/week	*Reference*			*Reference*			*Reference*		
3–4 days/week	0.8	0.6–1.1	0.110	0.9	0.7–1.2	0.325	0.8	0.6–1.0	0.063
≥5 days/week	0.6	0.4–0.8	**0.002**	0.6	0.4–0.8	**>0.001**	0.6	0.4–0.8	**<0.001**
**Cake/doughnuts intake**									
1–2 days/week	*Reference*			*Reference*			*Reference*		
3–4 days/week	0.8	0.6–1.0	0.092	0.6	0.5–0.9	**0.003**	0.6	0.5–0.8	**0.001**
≥5 days/week	0.6	0.4–0.8	**0.001**	0.7	0.5–0.9	**0.010**	0.6	0.4–0.8	**<0.001**

^a^ Normal weight as reference; ^b^ WC lower than the cutoff for age and genders (Taylor et al., 2000); ^c^ WHtR ≥0.5; OR = odds ratio (adjusted for age, nationality); CI = confidence interval; ^d^ active ≥1680 METs-min/week, inactive <1680 METs-min/week. Bold indicated significant association between variables.

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
