# Peer review of "The Relationship between Lifestyle Factors and Obesity Indices among Adolescents in Qatar"

_ijerph, 2019, doi:10.3390/ijerph16224428_

Round 1
Reviewer 1 Report
The authors report on “Relationship between Life Style factors and obesity indices among adolescent in Qatar”.
In the Abstract, the authors state that the sample data were collected using a “multi-stage sampling technique.” However, nowhere else is this important fact mentioned again or considered in this study. It’s not mentioned in the methods, including the statistical analysis and sample, nor the limitations of the study.
This is important because a multi stage sampling technique is an example of a complex survey design. Complex survey designs should not be analyzed in the same way as random samples. Most parametric statistics (of the type used by these authors) assume a random sample. Multi-stage samples usually are not random samples. To analyze a complex sample generally requires adjustments for variance issues and also for weighting issues. Parameter estimates of complex samples are generally different from those using random sampling. Thus the parameter estimate and generalizations of this study are somewhat questionable.
This paper has numerous spelling, punctuation, and grammatical errors that require a further English editing process.
The authors state (Page 2) that a validated questionnaire was used. They neither provide a reference for this nor state how and in which group this questionnaire was validated. Was it validated in Qataris or in another Gulf Arab group or in Europeans?
On Page 3 the authors state that 21 females who were included in the original sample did not have complete data and were not included in their final data analysis. Did this sample reduction have any impact on either power or their results? Were those with incomplete data similar to those with complete data generally (age, income, class level), etc. or were there systematic or other differences between them?
All of the Tables could be given better, more descriptive names. As an example, for Table 1 “Summary statistics on male and female Qatari adolescents” or some such title. Tables should be understandable without reference to the text.
Page 4 (but throughout the results section) of the manuscript the authors have a pattern of making statements like this (page 4). “No significant differences were noted between genders.” It would be clearer to the reader for example, if the authors said “No significant BMI or weight differences were noted between genders.” Then it would be clearer what exactly they are referring to with their standalone sentence.
The layout of Table 2 is very confusing and needs to be reworked. In column 3 of the physical activity variable, for example, one would find the following numbers in rows. 185, 33.1, 374, and 66.9. Obviously, the 33.1 and the 66.9 should be in a separate column.
In table 3 the N’s are not listed.
In the Discussion on Page 11, it would have seemed to me the authors would have compared their results with greater benefit to mostly other Gulf countries such as Kuwait, UAE, Oman, Bahrain, Saudi Arabia, etc. The ethno-history, genetics, diet, culture, lifestyles, geography, religious practices, ambient temperatures, and environments are very different in Qataris compared to adolescents in Sweden and Spain, etc. Similar obesity studies, particularly in adults, have been carried out and reported from other Gulf countries.
Presumably the Arab Teens Lifestyle Study sought and obtained human subjects permissions. This is not mentioned in this paper. Did the larger study or this one have human subjects clearance?
The authors mention that the study had some limitations. One that they mention is the cross-sectional nature of the study. How specifically is that a limitation? They don’t explain this. What are the implications for this study?
As I mentioned earlier, they never explain how variance and weighting issues were handled, if at all, in their analysis.
Author Response
Dear Editor,
On behalf of all authors of this manuscript, we would like to thank and acknowledge the efforts from you and the reviewers for putting adequate time to help us improve this manuscript. Please find below point by point response to comments.
Corresponding author,
|
Reviewer 1 |
|
|
Comment |
response |
|
In the Abstract, the authors state that the sample data were collected using a “multi-stage sampling technique.” However, nowhere else is this important fact mentioned again or considered in this study. It’s not mentioned in the methods, including the statistical analysis and sample, nor the limitations of the study
|
The sampling procedure was described in details and published previously
Details about the sampling method was added – page 2 line 32-36
A multistage, stratified, random sampling technique was used to select the sample. In the first stage, schools were listed separately and numbered from 1 to 24 for male and 1 to 23 for female. Based on the average number of students to be included from each school an interval of 2 was identified and then the schools were systematically selected as a multiplication of number 2, while the first school was selected randomly from the lists of both male’s and female’s schools. The total number of schools included in the study was 23 (11 male schools and 12 female schools). At the second stage, classes were chosen, using a random sampling method. At this stage, one class was randomly selected in each of the three grades (Grade 10, Grade 11 and Grade 12). The total number of classes selected was thus 69 classes
Kerkadi, A.; Hassan, A.S.; Al Thani, A.M.; Al Chetachi, W.; Akram, H.; Bawadi, H.; Vinodson, B.; Risk, M.R.N. Prevalence of general and abdominal obesity among adolescents attending independent schools in Qatar. Nut Food Sci 2018, 49(4), 687-699. doi.org/10.1108/NFS-09-2018-0260.
|
|
This is important because a multi stage sampling technique is an example of a complex survey design. Complex survey designs should not be analyzed in the same way as random samples. Most parametric statistics (of the type used by these authors) assume a random sample. Multi-stage samples usually are not random samples. To analyze a complex sample generally requires adjustments for variance issues and also for weighting issues. Parameter estimates of complex samples are generally different from those using random sampling. Thus the parameter estimate and generalizations of this study are somewhat questionable.
|
We fully agree with the reviewer that subjects were selected using Multi-Stage Cluster sampling scheme and data were analyzed ignoring the nested cluster structure. As such analysis assuming independence is questionable. Yes, currently data are analyzed as thought they were all selected randomly across the study population, ignoring the nested structure. We considered multi-level modelling approach to accounted for nested cluster structure but because all schools considered in this study were public schools, there ICC was negligible. This was expected because schools were homogenous in terms of the students socio-economic status. The so our statistician advised that when the ICC is almost closer to zero the estimates accounting for the nested structure will give similar estimates as we have presented. Add to this our major research question was to investigate the associations between lifestyle factors and obesity rather than estimating prevalence. |
|
This paper has numerous spelling, punctuation, and grammatical errors that require a further English editing process.
|
The manuscript was sent to a professional proof reading company ; certificate is provided |
|
The authors state (Page 2) that a validated questionnaire was used. They neither provide a reference for this nor state how and in which group this questionnaire was validated. Was it validated in Qataris or in another Gulf Arab group or in Europeans? |
The study is part of a big study in which the questionnaires were tested and validated in several Arab countries. This was clarified in the manuscript- page 2; line 26-19 page 3 line 3,4 1. Al-Hazzaa, H.M.; Musaiger, A.O. Arab Teens Lifestyle Study (ATLS): objectives, design, methodology and implications. Diabetes Metab Syndr Obes 2011, 4, 417-426, doi:10.2147/dmso.S26676. 2. Al-Hazzaa, H.M.; Al-Sobayel, H.I.; Musaiger AO. Convergent validity of the Arab Teens Lifestyle Study (ATLS) physical activity questionnaire. Int J Environ Res Public Health. 2011 ;8(9):3810-20. doi: 10.3390/ijerph8093810. 3. Musaiger, A. O., Bader, Z., Al-Roomi, K., & D'Souza, R. Dietary and lifestyle habits amongst adolescents. Food & nutrition research. 2011; 55, 7122. doi:10.3402/fnr.v55i0.7122 |
|
In Page 3 the authors state that 21 females who were included in the original sample did not have complete data and were not included in their final data analysis. Did this sample reduction have any impact on either power or their results? Were those with incomplete data similar to those with complete data generally (age, income, class level), etc. or were there systematic or other differences between them? |
The sample reduction does not have any impact on results. There were no differences between the two groups ( incomplete and complete data)
|
|
All of the Tables could be given better, more descriptive names. As an example, for Table 1 “Summary statistics on male and female Qatari adolescents” or some such title. Tables should be understandable without reference to the text.
|
Titles of the tables were revised. |
|
Page 4 (but throughout the results section) of the manuscript the authors have a pattern of making statements like this (page 4). “No significant differences were noted between genders.” It would be clearer to the reader for example, if the authors said “No significant BMI or weight differences were noted between genders.” Then it would be clearer what exactly they are referring to with their standalone sentence.
|
Thanks for the suggestion- results were revised and edited accordingly Page 3, line 30-41 |
|
The layout of Table 2 is very confusing and needs to be reworked. In column 3 of the physical activity variable, for example, one would find the following numbers in rows. 185, 33.1, 374, and 66.9. Obviously, the 33.1 and the 66.9 should be in a separate column.
|
Table 2 was edited accordingly Page 4, page 5
|
|
In table 3 the N’s are not listed |
Added |
|
In the Discussion on Page 11, it would have seemed to me the authors would have compared their results with greater benefit to mostly other Gulf countries such as Kuwait, UAE, Oman, Bahrain, Saudi Arabia, etc. The ethno-history, genetics, diet, culture, lifestyles, geography, religious practices, ambient temperatures, and environments are very different in Qataris compared to adolescents in Sweden and Spain, etc. Similar obesity studies, particularly in adults, have been carried out and reported from other Gulf countries. |
We did a comparison with data from Middle East countries including GCC. Most the selected Arabic countries have used the ATLS instrument. Most of Arabic countries are sharing same culture and habits I agree with your comments regarding the comparison with Europeans countries. We wanted to compare our results with others from another environment and socio economic status |
|
Presumably the Arab Teens Lifestyle Study sought and obtained human subjects permissions. This is not mentioned in this paper. Did the larger study or this one have human subjects clearance? |
The study proposal has been reviewed by the ministry of Public Health and ministry of education review board. A statement about the study approval was added in the manuscript. Page 2, line 39-42 |
|
The authors mention that the study had some limitations. One that they mention is the cross-sectional nature of the study. How specifically is that a limitation? They don’t explain this. What are the implications for this study? |
This study can not infer any causal relationships based on the cross-sectional design. Limitations have been updated and a paragraph on implications was added in conclusion section Page 13, line 32 |
|
As I mentioned earlier, they never explain how variance and weighting issues were handled, if at all, in their analysis
|
|
Reviewer 2 Report
The purpose of the study is to evaluate the relationship between general and abdominal obesity, and lifestyle factors among adolescents in Qatar. There are some merits of the study such as using a large sample, but many aspects of the manuscript are not clear. My comments are detailed below, and I hope they could be useful.
Abstract
IOTF, WHrR are not defined
The results section in the abstract discusses the prevalence, but conclusion sentence discusses the relationship between low intake of breakfast and other. That is inconsistent.
Introduction
It is generally clear, but the researcher(s) need to make it clear what lifestyle factor is and what are included in the lifestyle factor. Similarly, the researcher (s) should be clear about the type of body composition that is used.
p2, L4-7, this paragraph does is very broad. I suggest the authors using the space to discuss about the lifestyle factors
Methods
ATLS and later paragraph described elsewhere is not sufficient. The researcher(s) should briefly describe that, then maybe using the term described elsewhere.
2.3 What is the validity and reliability of the questionnaire? What types of the questions were asked?
2.4 The cutoff score 1680 met-min/week seems to be low compared to the commonly used 3000 met-min/week.
2.7 The statistical tests (t and λ2?) were used to assess the differences in what variables between sexes? Was the sampling plan and weighting considered in the logistic regression analysis?
Results
It is not clear what statistical analyses were used for Table 1 results. WHrR: 0.52 vs 0.50, p < .001?
Additionally, effect sizes should be reported.
Table 2 is hard to follow. Not clear what statistical analysis was used for significance testing.
Table 4,5,6 and remaining sections: I recommend the researcher(s) briefly review the difference and similarity between BMI, WHrR, and WC and select one variable for this research as they are similar instruments measuring body composition. Having three relatively similar measures complicates the statistical analysis needed, leads to unnecessarily large and confusing tables, and muddies the discussion.
Author Response
Dear Editor,
On behalf of all authors of this manuscript, we would like to thank and acknowledge the efforts from you and the reviewers for putting adequate time to help us improve this manuscript. Please find below point by point response to comments.
Corresponding author,
|
Reviewer 2 |
|
|
Comment |
Response |
|
Abstract :IOTF, WHrR are not defined
|
IOTF and WHtR were defined in the abstract Page 1, line 20, line 22
|
|
Abstract :The results section in the abstract discusses the prevalence, but conclusion sentence discusses the relationship between low intake of breakfast and other. That is inconsistent. |
Abstract was modified according to the reviewer’s comment Page 1; line 26-30 |
|
It is generally clear, but the researcher(s) need to make it clear what lifestyle factor is and what are included in the lifestyle factor. Similarly, the researcher (s) should be clear about the type of body composition that is used. |
Clarification was added Page 2, line 5
|
|
p2, L4-7, this paragraph does is very broad. I suggest the authors using the space to discuss about the lifestyle factors |
Paragraph was modified Page 2, line 3-5 |
|
Methods: ATLS and later paragraph described elsewhere is not sufficient. The researcher(s) should briefly describe that, then maybe using the term described elsewhere. |
Details are added Page 2 line 26-28 Page 2 line 33-37
|
|
What is the validity and reliability of the questionnaire? What types of the questions were asked? |
The study is part of a big study in which the questionnaires were tested and validated in several Arab countries. This was clarified in the manuscript- page 2; line 26-19 page 3 line 3,4 1. Al-Hazzaa, H.M.; Musaiger, A.O. Arab Teens Lifestyle Study (ATLS): objectives, design, methodology and implications. Diabetes Metab Syndr Obes 2011, 4, 417-426, doi:10.2147/dmso.S26676. 2. Al-Hazzaa, H.M.; Al-Sobayel, H.I.; Musaiger AO. Convergent validity of the Arab Teens Lifestyle Study (ATLS) physical activity questionnaire. Int J Environ Res Public Health. 2011 ;8(9):3810-20. doi: 10.3390/ijerph8093810. 3. Musaiger, A. O., Bader, Z., Al-Roomi, K., & D'Souza, R. Dietary and lifestyle habits amongst adolescents. Food & nutrition research. 2011; 55, 7122. doi:10.3402/fnr.v55i0.7122 |
|
The cutoff score 1680 met-min/week seems to be low compared to the commonly used 3000 met-min/week. |
We used the cutt of 1680 METs-min/week which is equivalent to 1 hour of moderate physical activity per day. It is considered as the minimum amount of PA required per day. The decision was made for all other atlas studies- we used same cut-off points to have to make the results comparable. |
|
The statistical tests (t and λ2?) were used to assess the differences in what variables between sexes? Was the sampling plan and weighting considered in the logistic regression analysis? |
Sample weight was not calculated because all the school we worked with were very homogenous. All school were small public school were students socioeconomic backgrounds are very similar. After consultation with our statistician, he suggested that results will not change the results; especially that our research questions are about associations rather than estimating prevalences. |
|
Results: It is not clear what statistical analyses were used for Table 1 results. WHrR: 0.52 vs 0.50, p < .001?
|
Thanks for noticing this. We check the analysis and P<0.001 |
|
Table 2 is hard to follow. Not clear what statistical analysis was used for significance testing. |
Clarified |
|
Table 4,5,6 and remaining sections: I recommend the researcher(s) briefly review the difference and similarity between BMI, WHrR, and WC and select one variable for this research as they are similar instruments measuring body composition. Having three relatively similar measures complicates the statistical analysis needed, leads to unnecessarily large and confusing tables, and muddies the discussion. |
We thank the reviewer for this suggestion. We, however prefer to keep three indicators to present general and abdominal obesity. |
Reviewer 3 Report
Dear Editor,
This is a well written manuscript that focuses on determining the influence of physical inactivity, sedentary behaviors (including screen time), and unhealthy diet on being overweight/and or obese in adolescents from Qatar.
The manuscript does provide a substantive review of the contemporary literature that justifies the research questions. The manuscript is organized with appropriate material in the other sections. The topic of the manuscript fits well within the scope of the journal. However, the relevance of the clinical question addressed in the current manuscript is limited.
The manuscript proposes to address a question that can assist in clinical decision making for designing healthier diet and physical activity programs for the youth in Qatar. However, the current manuscript is limited by its design and analysis.
Methods: The main focus of the manuscript is centered on determining factors that influence obesity and weight in adolescents in Qatar.
Abdominal obesity variable is not well defined. Is WC or BMI or Waist to height ratio or all three used to define this variable. Please clarify wth rationale and citations to support your statements. The questionnaire listed under 2.3 needs information on the type of validity reported with citation. The factors related to diet are one dimensional and only focus on western diet. Other food items that are and would qualify as greasy/unhealthy, should also be included to ensure the construct for the variable “diet” is defined and upheld. Also, the most recent guidelines for screen time do not specify 2 hours/day. Please verify
Data analysis: The authors provide details about the statistical methods that were employed in the current study.
Results: This will need to be reviewed after the limitations cited in the methods sections are clarified by the authors.
Discussion: This will need to be reviewed after the limitations cited in the methods sections are clarified by the authors.
Also, include influence of cultural differences in gender that might impact physical activity and other lifestyle behaviors.
Author Response
Dear Editor,
On behalf of all authors of this manuscript, we would like to thank and acknowledge the efforts from you and the reviewers for putting adequate time to help us improve this manuscript. Please find below point by point response to comments.
Corresponding author,
|
Reviewer 3 |
|
|
Comment |
Response |
|
Abdominal obesity variable is not well defined. Is WC or BMI or Waist to height ratio or all three used to define this variable. Please clarify with rationale and citations to support your statements. |
Abdominal obesity was defined by WHtR and WC . Details have been updated in the section2.2
|
|
The questionnaire listed under 2.3 needs information on the type of validity reported with citation |
The study is part of a big study in which the questionnaires were tested and validated in several Arab countries. This was clarified in the manuscript- page 2; line 26-19; page 3 line 3,4 Al-Hazzaa, H.M.; Musaiger, A.O. Arab Teens Lifestyle Study (ATLS): objectives, design, methodology and implications. Diabetes Metab Syndr Obes 2011, 4, 417-426, doi:10.2147/dmso.S26676. Al-Hazzaa, H.M.; Al-Sobayel, H.I.; Musaiger AO. Convergent validity of the Arab Teens Lifestyle Study (ATLS) physical activity questionnaire. Int J Environ Res Public Health. 2011 ;8(9):3810-20. doi: 10.3390/ijerph8093810. Musaiger, A. O., Bader, Z., Al-Roomi, K., & D'Souza, R. Dietary and lifestyle habits amongst adolescents. Food & nutrition research. 2011; 55, 7122. doi:10.3402/fnr.v55i0.7122 |
|
The factors related to diet are one dimensional and only focus on western diet. Other food items that are and would qualify as greasy/unhealthy, should also be included to ensure the construct for the variable “diet” is defined and upheld.
|
There is evidence that dietary habits of adolescents in the gulf countries is highly influenced by the western diet (Musaiger and Al Hazzaa, 2012; Aboul Enein et al, 2017; Golzarand et al, 2012). The dietary questionnaire was developed , validated and tested among adolescents in many Arabic countries. Arabic fast food dishes (shish touk, falafel, shawarma ..) consumed by adolescents were included in the fast food group.
|
|
Also, the most recent guidelines for screen time do not specify 2 hours/day. Please verify
|
I agree that the recent APA guidelines for screen time does not specify time. APA sets time only for children aged 2-5 years (1 hour/day). We used the cutoff of less than 2 h to be able to compare with other studies done in the region and outside. |
|
Results: This will need to be reviewed after the limitations cited in the methods sections are clarified by the authors. |
Done
|
|
Discussion: This will need to be reviewed after the limitations cited in the methods sections are clarified by the authors. |
Revised
|
|
Also, include influence of cultural differences in gender that might impact physical activity and other lifestyle behaviors. |
A paragraph listing cultural barriers to physical activity and life style behaviors are added in the text
|
Reviewer 4 Report
This study examines the relationship between lifestyle factors (e.g., physical activity, dietary, sedentary) and obesity (general and abdominal) in adolescents in Qatar. Due to the lack of research examining this area of interest among adolescents in Qatar, it provides unique insight into determining what lifestyle factors are the most crucial for preventative and secondary treatment to addressing Qatar's current obesity problem.
However, there are several issues with the manuscript in its current form:
Throughout the manuscript are errors with use of the English language. It is commonly "lifestyle" as one word and not separated as two. There are several oddly used ways of describing past research (e.g., "different researches"; Line 5; Page 2). Several words are randomly capitalized (e.g., Obesity; Line 4; Page 2) when they shouldn't be. There are overlooked errors (e.g., 3days; Line 15; Page 3). All incorrect usages of the English language need to be addressed as it disrupts the reading of the manuscript. Consistency is needed whether to use "gender" or "sex". What is this validated questionnaire that was used to collect lifestyle behavior data? If it's a validated questionnaire, please provide the citation and/or validity data. For physical activity, did the questionnaire only ask about duration and intensity of the various physical activities? What about frequency? In the Discussion there is no discussion of what these results/findings mean and how they will be utilized to address pediatric obesity issue currently seen in Qatar. What do interventions need to target based on the results of this study?
Although this study is of great interest and a positive contribution to the scientific field of pediatric obesity, especially for Qatar, the manuscript requires major revision.
Author Response
Dear Editor,
On behalf of all authors of this manuscript, we would like to thank and acknowledge the efforts from you and the reviewers for putting adequate time to help us improve this manuscript. Please find below point by point response to comments.
Corresponding author,
|
Reviewer 4 |
|
|
Comment |
Response |
|
Throughout the manuscript are errors with use of the English language |
The manuscript was sent to a professional proof reading company ; certificate is provided |
|
It is commonly "lifestyle" as one word and not separated as two. |
Corrected in the manuscript
|
|
There are several oddly used ways of describing past research (e.g., "different researches"; Line 5; Page 2).
|
Word different was removed and replaced by another word
|
|
Several words are randomly capitalized (e.g., Obesity; Line 4; Page 2) when they shouldn't be. There are overlooked errors (e.g., 3days; Line 15; Page 3). |
Corrected
|
|
All incorrect usages of the English language need to be addressed as it disrupts the reading of the manuscript.
|
The manuscript was sent to a professional proof reading company ; certificate is provided |
|
Consistency is needed whether to use "gender" or "sex". |
Sex has been changed to gender in the whole text |
|
What is this validated questionnaire that was used to collect lifestyle behavior data? If it's a validated questionnaire, please provide the citation and/or validity data. |
See page 2; line 26-19 page 3 line 3,4 |
|
For physical activity, did the questionnaire only ask about duration and intensity of the various physical activities? What about frequency? |
The questionnaire asked about duration, intensity and frequency. More details are provide in section 2.4
|
|
In the Discussion there is no discussion of what these results/findings mean and how they will be utilized to address pediatric obesity issue currently seen in Qatar. What do interventions need to target based on the results of this study? |
A paragraph covering recommendations based on the research findings was added in the section 5. Line 27-28.
|
Round 2
Reviewer 4 Report
Thank you for correcting/addressing my previous suggestions/comments. I have no further suggestions or comments and approve the manuscript in its present form. This manuscript will be of great contribution to the field of Pediatric Obesity especially for the people of Qatar.